# A Proteomics-Based Approach Identifies the NEDD4 Adaptor NDFIP2 as an Important Regulator of Ifitm3 Levels

**DOI:** 10.3390/v15101993

**Published:** 2023-09-26

**Authors:** Federico Marziali, Yuxin Song, Xuan-Nhi Nguyen, Lucid Belmudes, Julien Burlaud-Gaillard, Philippe Roingeard, Yohann Couté, Andrea Cimarelli

**Affiliations:** 1Centre International de Recherche en Infectiologie (CIRI), Univ Lyon, Inserm, U1111, Université Claude Bernard Lyon 1, CNRS, UMR5308, ENS de Lyon, 69100 Lyon, Francexuan-nhi.nguyen@ens-lyon.fr (X.-N.N.); 2Université Grenoble Alpes, INSERM, CEA, UA13 BGE, CNRS, CEA, 38000 Grenoble, France; lucid.belmudes@cea.fr (L.B.); yohann.coute@cea.fr (Y.C.); 3Plateforme IBiSA de Microscopie Electronique, Université de Tours et CHU de Tours, 37000 Tours, France; julien.gaillard@univ-tours.fr (J.B.-G.); roingeard@med.univ-tours.fr (P.R.); 4INSERM U1259, Université de Tours et CHU de Tours, 37000 Tours, France

**Keywords:** IFITM3, virus, NEDD4, NDFIP2, interferon

## Abstract

IFITMs are a family of highly related interferon-induced transmembrane proteins that interfere with the processes of fusion between viral and cellular membranes and are thus endowed with broad antiviral properties. A number of studies have shown how the antiviral potency of IFITMs is highly dependent on their steady-state levels, their intracellular distribution and a complex pattern of post-translational modifications, parameters that are overall tributary of a number of cellular partners. In an effort to identify additional protein partners involved in the biology of IFITMs, we devised a proteomics-based approach based on the piggyback incorporation of IFITM3 partners into extracellular vesicles. MS analysis of the proteome of vesicles bearing or not bearing IFITM3 identified the NDFIP2 protein adaptor protein as an important regulator of IFITM3 levels. NDFIP2 is a membrane-anchored adaptor protein of the E3 ubiquitin ligases of the NEDD4 family that have already been found to be involved in IFITM3 regulation. We show here that NDFIP2 acts as a recruitment factor for both IFITM3 and NEDD4 and mediates their distribution in lysosomal vesicles. The genetic inactivation and overexpression of NDFIP2 drive, respectively, lower and higher levels of IFITM3 accumulation in the cell, overall suggesting that NDFIP2 locally competes with IFITM3 for NEDD4 binding. Given that NDFIP2 is itself tightly regulated and highly responsive to external cues, our study sheds light on a novel and likely dynamic layer of regulation of IFITM3.

## 1. Introduction

Interferon-induced transmembrane proteins (IFITMs) are a family of highly related membrane-bound proteins involved in viral control and bone metabolism (IFITMs 1, 2, 3 and 5) [1,2]. More specifically, IFITMs 1, 2 and 3 are known to inhibit the replication of numerous viruses by interfering with their ability to fuse with cellular membranes. This occurs at two distinct moments of the virus life cycle: during target cell infection, when IFITMs trap incoming virion particles in endo/lysosomes, and during the production of new virion particles from infected cells, where IFITMs lead to the production of virions with a membrane fusogenicity defect [3,4,5].

The changes in membrane behavior that underlie the antiviral effects of IFITMs are linked to the rigidification of the membranes of cells expressing IFITMs. At present, it remains unclear whether such rigidification is the direct result of the insertion of IFITMs in membranes themselves, or whether it is an indirect consequence of changes driven by IFITMs in their lipid composition, as evidence in support of both possibilities is present in the literature [6,7,8,9,10].

Given that they act as modulators of the behavior of membranes, it is not surprising that the intracellular levels of IFITMs are tightly regulated. While a first layer of regulation of IFITMs is transcriptional and relies on the presence of interferon, a second one occurs post-translationally. IFITMs have been described to be methylated, ubiquitinated, phosphorylated and palmitoylated by the methylase and demethylases SET7/LSD1 [11,12], the Fyn kinase [13], the NEDD4 E3 ubiquitin ligase [14] and the palmitoyltransferase ZDHHC20 [15]. The interdependence of such changes has not been explored to its full extent and will likely constitute the next frontier in the study of the biology of IFITMs. For example, phosphorylation on tyrosine at position 20 has been shown to modulate the extent of IFITM3 ubiquitination [13], and it can be hypothesized that some modifications may act competitively on each other when they occur on the same residues, as is the case for ubiquitination and mono-methylation, both of which occur on lysines.

Besides being the target of several enzymes, IFITMs also associate with protein cofactors that regulate their intracellular distribution and thus also contribute to the regulation of the steady-state levels and availability of the different IFITMs, as shown for the adaptor proteins 2 and 3 (AP2/AP3) [16,17]. On the whole, the proteome associated with IFITM proteins influences many key parameters in their biology, and, hence, it bears the possibility of influencing the efficiency with which IFITMs can act against different viruses.

To identify novel functional partners of IFITM3, we took advantage of the fact that, in the absence of viral infection, this protein can be incorporated into exosomal vesicles [18,19]. We thus hypothesized that IFITM3 cofactors could be piggybacked into extracellular vesicles simply by virtue of protein–protein interaction and could be thus identified using mass spectrometry (MS)-based proteomics. IFITM3 packaging into extracellular vesicles was therefore used here as a method to concentrate potential cofactors in a manner that is conceptually similar to co-immunoprecipitations. With respect to the latter, however, this strategy offers the advantage of avoiding detergent treatment of cell lysates, which is instead required for immunoprecipitations and, by extracting membrane-embedded proteins from their original environment, may lead to artefacts.

To this end, we purified the exosomes produced from cells expressing or not expressing IFITM3 and performed an MS-based proteomic analysis. Among the proteins specifically enriched in IFITM3 exosomes, we focused on the NEDD4-interacting family protein 2 (NDFIP2) [20,21,22]. NEDD4 is an E3 ubiquitin ligase that has been described by the Yount lab as an important regulator of the steady-state levels of IFITM3 [14], leading us to further explore the role of NDFIP2 in the relationship between this enzyme and IFITM3. NDFIP2 is a protein that acts as a general membrane adaptor/regulator of members of the homonymous subfamily of E6AP C-terminus E3 ubiquitin ligases (HECT) that play critical roles in numerous physiological processes [23]. In particular, NDFIP2 normally acts to recruit otherwise cytoplasmic HECT members on membranes, and it has been described to be able to either stimulate or depress their activity on specific targets [21]. Using CRISPR/Cas9 knockout cells, as well as overexpression experiments, our data indicate that NDFIP2 is a positive regulator of IFITM3 and that, despite the fact that it is important to recruit NEDD4 and IFITM3 on lysosomes, it likely works locally by diverting NEDD4 from IFITM3. As a result, A549 cells, in which NDFIP2 has been genetically removed by CRISPR/Cas9, exhibit lower levels of IFITM3 and become highly susceptible to infection. Overall, our results identify NDFIP2 as an important factor in the biology of IFITM3. Given that NDFIP2 is itself exquisitely regulated through distinct cell activation cues, these findings add a novel layer of complexity in the regulation of the steady-state levels of IFITMs.

## 2. Materials and Methods

### 2.1. Cells, Plasmids, Antibodies and Compounds

Human embryonic kidney and human lung epithelial cells (HEK293T and A549, ATCC cat. CRL-3216 and CCL-185, respectively) were cultured in complete DMEM media with 10% Fetal Calf Serum (Sigma, Saint Louis, MI, USA cat. F7524). The DNA coding for Flag-tagged IFITM3 has been previously described [24]. pCI-HA-NEDD4 was a gift from Joan Massague (Addgene, Cambridge, MA, USA plasmid 27002), while HA-ubiquitin coding DNA was a gift from Pierre Jalinot (ENS-Lyon). Human Myc-tagged NDIP2 and PY_1,2,3_/F mutant coding plasmids were obtained from Thomas Mund and Hugh Pelham at the MRC Laboratory of Molecular Biology and Cell Biology Cambridge, UK [25]. In the PY_1,2,3_/F NDFIP2 mutant, the tyrosine residues in positions 151, 177 and 186 had been mutated to phenylalanine [25]. The target sequences for CRISPR/Cas9 gene inactivation were cloned in the context of the lentiviral vector (LentiCRISPRv2, Addgene plasmid 52961) and used to produce lentiviral vectors, as specified in the relevant section below. The two target sequences used for NDFIP2 were GAAAAGTAAGGCAAGTCCGG and AGCATGCTCAATAGCGCGCG. The non-targeting control sequence was CGAGGAGCTGTTCACCGGGG. The following oligos were used to amplify the genomic region flanking the targets mentioned above: NDFIP2crispchkFw: CCATCCCTTGTCAGCCTCTGT and NDFIP2crispchkRv: AGGGAGTCTTCTCCGTGCTCAG. PCR products were cloned, and eight to ten individual clones were sequenced to visualize the genetic deletion.

The following primary antibodies were used for WB or confocal microscopy, as indicated. Mouse monoclonal antibodies: anti-Flag, anti-α-Tubulin, anti-HA and anti-VSV-G (Sigma, Saint Louis, USA cat. F3165, T5168, H3663 and V5507, respectively), anti-LAMP2 (Santa Cruz Biotechnology, Dallas, USA cat. sc-18822); anti-GM130 (BD biosciences, Franklin Lakes, NJ, USA cat. 610823); anti-myc (Eurogentec, Seraing, Belgium cat. MMS-150R-0200). Rabbit polyclonal antibodies: anti-IFITM3 (Proteintech, Rosemont, IL, USA cat. 11714-1-AP), anti-GM130 (Abcam, Cambridge, UK cat. ab52649), anti-Flag and anti-HA (Sigma, Saint Louis, MI, USA cat. F7425 and H6908). Goat polyclonal anti-Flag antibody (Abcam, Cambridge, UK cat. ab1257).

The following secondary antibodies were used for WB: anti-mouse, anti-rabbit IgG-peroxidase conjugated (Sigma, Saint Louis, USA cat. A9044 and AP188P). The following ones were used for confocal microscopy: donkey anti-rabbit IgG conjugated with Alexa Fluor 594 or 647 or 555, donkey anti-mouse IgG–Alexa Fluor 488 conjugate and donkey anti-Goat IgG- Alexa Fluor 594 (Life Technologies, Carlsbad, USA, cat. A-21207, or A-32795 or A-32974, A-21202 and A-11058).

CRISPR/Cas9 cells were selected and maintained as a pool with puromycin (Sigma, Saint Louis, MI, USA cat. P8833). Cells were discarded routinely after one month in cell culture to avoid the possible drifting of the lineage. Doubling time and cell viability were assessed via cell counting and trypan blue exclusion.

### 2.2. Preparation of Exosomal Fractions for Proteomic Analysis

Forty 10 cm well plates were ectopically transfected using calcium phosphate with control or IFITM3-expressing DNAs (40 μg/DNA per plate) overnight. Media were replaced the morning after, and the exosomal fraction was harvested twenty-four hours later from the cell supernatant. The supernatant was first syringe-filtered (0.45 μm), layered onto a two-step sucrose cushion (45/25% sucrose *w*/*v*, in 10 mM Tris/Cl pH 7.5, 100 mM NaCl and 1 mM EDTA) and ultracentrifuged (25.000 rpm, 2 h). The interface between the 45 and 25% sucrose was harvested, diluted three times in PBS and purified again using ultracentrifugation over a single layer of 25% sucrose (25.000 rpm, 2 h). Exosomes were then harvested at the bottom of the tube.

### 2.3. Mass-Spectrometry-Based Proteomic Analyses

The proteins present in the purified exosomes were solubilized in Laemmli buffer and stacked on the top of a 4–12% NuPAGE gel (Thermo Fisher Scientific-Invitrogen, Waltham, MA, USA). After staining with R-250 Coomassie Blue (Biorad, Hercules, CA, USA) or silver staining (Thermo Fisher Scientific-Pierce, Waltham, MA, USA), proteins were digested in-gel using trypsin (modified, sequencing purity, Promega, Madison, WI, USA), as previously described [26]. The resulting peptides were analyzed using online nanoliquid chromatography coupled to MS/MS (Ultimate 3000 and LTQ-Orbitrap Velos Pro, Thermo Fisher Scientific, Waltham, MA, USA) using a 120 min gradient. For this purpose, the peptides were sampled on a precolumn (300 μm × 5 mm PepMap C18, Thermo Fisher Scientific, Waltham, MA, USA) and separated in a 75 μm × 250 mm C18 column (PepMap C18, 3 μm, Thermo Fisher Scientific, Waltham, MA, USA). The MS and MS/MS data were acquired using Xcalibur (Thermo Fisher Scientific, Waltham, MA, USA).

Peptides and proteins were identified and quantified using MaxQuant (version 1.5.3.30, [27]), the UniProt database (Homo sapiens taxonomy, 20161026 download) and the frequently observed contaminant database embedded in MaxQuant. Trypsin/P was chosen as the enzyme, and 2 missed cleavages were allowed. The peptide modifications allowed during the search were as follows: carbamidomethylation (C, fixed), acetyl (Protein N-ter, variable) and oxidation (M, variable). The minimum peptide length was set to seven amino acids. The maximum false discovery rate—calculated by employing a reverse database strategy—was set to 0.01 at peptide and protein levels. The Match Between Run option was enabled with the default settings. Protein intensities were calculated from the extracted MS intensities of unique and razor peptides and used to compute iBAQ values [28].

Statistical analyses were performed using ProStaR [29] on extracted iBAQ values for each protein. The proteins identified in the reverse and contaminant databases, proteins only identified by site, proteins identified with only one peptide and proteins exhibiting less than three iBAQ values in one condition were discarded from the list. After log2 transformation, iBAQ values were normalized via median centering before missing value imputation (replacing missing values by the 2.5 percentile value of each column). Statistical testing was then conducted using the limma *t*-test. Differentially expressed proteins were sorted out using a log2 (fold change) cut-off of 2.5 and a *p*-value < 0.005, allowing for a false discovery rate below 1% to be reached, as calculated using the Benjamini–Hochberg method.

### 2.4. Immuno-Gold Electron Microscopy

Nickel grids coated with Formvar/carbon were deposited on a drop of purified, unfixed samples for five minutes prior to sequential incubation with an anti-Flag antibody (F7425, Sigma, St-Louis, MO, USA), followed by incubation with a 1:30 gold-conjugated (10 nm) goat-anti-Rabbit IgG (Aurion, Wageningen, The Netherlands) and fixation in 1% glutaraldehyde. Negative staining was performed using 2% uranyl acetate (Agar Scientific, Stansted, UK) prior to transmission electron microscope analyses (JEOL 1011, Tokyo, Japan).

### 2.5. Confocal Microscopy

Cells were directly seeded on glass coverslips coated with poly-L-Lysine 0.01% (Sigma, St-Louis, MO, USA cat. P4832) after Lipofectamine 3000-mediated DNA transfection, according to the manufacturer’s instructions (ThermoFisher-Invitrogen, Waltham, MA, USA cat. L3000008). Twenty-four hours later, the cells were fixed with PFA (3.7%), then permeabilized with 0.1% Triton X-100 in PBS and then incubated with the indicated primary and secondary antibodies. The coverslips were stained with a DAPI-containing solution (ThermoFisher, Waltham, MA, USA cat. 62248) and mounted using the anti-quenching solution Fluormount G (Southern Biotech, Birmingham, AL, USA). Fluorescent confocal images were acquired on a Zeiss LSM 880 AiryScan confocal microscope. The pictures were analyzed with ImageJ software.

### 2.6. Transfections and Immunoprecipitations

HEK293T cells were transfected with DNAs coding for the indicated proteins using calcium phosphate at a 1:1:0.2 ratio for NEDD4-, IFITM3- and NDFIP2-specific DNA. Two days afterwards, the cells were lysed in 150 μL of lysis/IP buffer (50 mM Tris/HCl pH 7.4, 150 mM NaCl, 0.1% NP-40, 1 mM EDTA), supplemented with a cocktail of protease inhibitors (Sigma, cat. 4693132001). Immunoprecipitations were carried out for two–three hours using 1 μg of the monoclonal antibodies indicated above, followed by an additional two hours incubation time with Protein A–sepharose beads (GE healthcare, cat. 17-5238-01). After extensive washing, samples were analyzed using WB. IP and washing steps were carried out using the same lysis/IP buffer mentioned above. For WB analyses, the cells were instead lysed directly in 100 μL of RIPA buffer (150mM NaCl, 1% NP40, 0.5% deoxycholate, 0.1% SDS, 50 mM TrisCl pH 8.0), supplemented with a cocktail of protease inhibitors (Sigma, cat. 4693132001), twenty-four hours after transfection. For ubiquitin experiments, DNA coding for HA-ubiquitin was added along with DNAs coding for IFITM3 or NDFIP2s at a ratio of 2:1:2, respectively. Cell lysis and IPs were then carried out as described above.

### 2.7. Densitometric Quantification

WB signals were acquired on a BioRad Molecular Imager Gel Doc apparatus, and densitometric quantifications of the specific WB bands were carried out using Imager lab software.

### 2.8. Vector Productions and Viral Infections

Lentiviral-mediated gene transduction was used to produce pools of cells with a deletion in ndfip2. HIV-1-based vectors were produced via the transient transfection of HEK293T cells with DNAs coding for the HIV-1 structural polyproteins Gag/Gag-Pro-Pol and the pantropic envelope VSV-G that allows for the infection of a broad spectrum of target cells, as well as two distinct mini-viral genomes (LentiCRISPRv2) that expressed one single-guide RNA specific for the gene of interest each along with the Cas9 enzyme (at a ratio of 8:4:4:4, respectively, for a total of 20 μg /10 cm plate). Forty-eight hours after transfection and media change, viruses were syringe-filtered (0.45 μm) and used directly for three rounds of gene transduction on target cells (one every day, for 3 days). Modified cells were selected thanks to the puromycin resistance marker present in the LentiCRISPRv2 genome.

A single round of infection-based HIV-1 lentiviral vectors were produced as described above, substituting the LentiCRISPRv2 mini-viral genome with a version expressing a CMV:GFP expression cassette [30].

Replication-competent GFP-coding VSV-Indiana serotype Rhabdovirus has been described before [31]. In both cases, the infectious titers of these viral preparations were measured after the challenge of HeLa cells with dilutions of the virus followed by the quantification of the number of infected cells three days post-infection in the case of HIV-1 and one day in the case of VSV. Infections in A549 cells were carried out similarly with multiplicities of infection (MOIs) ranging from 0.01 for VSV and 0.5 for HIV-1. Infectious titers were determined by performing infections with serial virus dilutions on target HEK293T cells, prior to a flow cytometry analysis of the percentage of infected cells, as described above.

### 2.9. Statistical Analyses

In addition to the specific statistical tests mentioned in the analysis of the MS results (see the relevant section above), two-tailed Student t or one-way Anova tests were performed using Prism 8 software, as specified in the legends of the figures.

## 3. Results

### 3.1. IFITM3 Exosome Proteomics to Identify Potential IFITM3 Cofactors

Several reports indicate that IFITM3 can be incorporated into exosomes [18,19]. We thus reasoned that the ectopic expression of IFITM3 could lead to vesicles enriched in potential IFITM3 cofactors, transported outside the cell into exosomes only by virtue of their physical interaction with IFITM3. Under this hypothesis, their presence in IFITM3 exosomes would provide a first indication of a potential protein–protein interaction, irrespectively of an active role in exosome biogenesis, or functions.

To this end, HEK293T cells were transfected with DNA coding for Flag-tagged human IFITM3 or a control vector, and forty-eight hours later, extracellular vesicles were purified from cell supernatants using two successive ultracentrifugation steps (first on a two-step sucrose gradient 25%/45% and the second through a single 25% sucrose cushion, *w*/*v*) prior to analysis. The proteins present in the purified exosomal fractions could be easily detected via SDS-PAGE gel/silver staining, and IFITM3 was strongly incorporated into the vesicles, as assessed using immuno-gold electron microscopy (Figure 1a). MS analysis identified and reliably quantified over 1000 proteins in the control and IFITM3 samples, 72 of which were differentially present in the IFITM3 condition, and, specifically, 68 were over- and 4 were under-represented by more than 5-fold (log2 fold change ≥2.5 or ≤2.5, and *p*-value ≤ 0.005, i.e., −log10 *p*-value = 2.3; Benjamini–Hochberg FDR < 1%) with respect to the control (Figure 1b,c, respectively; Appendix A).

Perhaps not surprisingly given the simple purification method used here, ribosomal proteins were abundant in both samples. However, their presence was higher in IFITM3 vesicles, which skewed the Ingenuity Pathway Analysis towards translation processes (Appendix A). This finding was not explored further, but it could be related to the recent reported effects of IFITM3 on protein translation [32].

Among the proteins found most enriched in IFITM3-containing vesicles, we decided to focus on Nedd4 Family Interacting Protein 2 (NDFIP2), given that this protein is important in the regulation of NEDD4 ([21] and Appendix A), an E3 ubiquitin ligase that has itself been found to be involved in IFITM3 biology [14].

### 3.2. NDFIP2 Regulates IFITM3 Levels

To determine the functional relevance of NDFIP2 on IFITM3 regulation, pools of control and NDFIP2-KO cells were generated in A549 cells following the lentiviral-mediated gene transduction of Cas9 and RNA guides. Transduced cells were then selected in pools thanks to the drug resistance genes present on the lentiviral vectors. The genetic removal of NDFIP2 was confirmed via sequencing, and it did not exert measurable effects on cell viability and growth, at least under the experimental conditions used here (Appendix A). Equal numbers of control and NDFIP2-KO cells were then lysed to measure the steady-state levels of IFITM3 following WB and densitometric quantification. Under these conditions, we found that IFITM3 levels were reduced in NDFIP2-KO cells, indicating that this protein contributes heavily to the steady-state accumulation of IFITM3 in the cells and that it may act to promote IFITM3 stability (Figure 2a). To test the hypothesis that NDFIP2 may titer out NEDD4 locally, diverting this E3 ubiquitin ligase away from its IFITM3 target, we co-expressed a constant level of IFITM3 along with increasing levels of NDFIP2 to saturate endogenous NEDD4 in HEK293T cells (Figure 2b).

Under these conditions, increasing levels of NDFIP2 led to an increased accumulation of IFITM3, lending support to the hypothesis that NDFIP2 may actually compete for binding to NEDD4 with IFITM3. Of note, the overexpression of NEDD4 along with a constant amount of IFITM3 did not lead to notable changes in the steady-state levels of the latter (Appendix A). This finding is in line with previous observations [14], and we believe that it may be due to the fact that NEDD4 levels are already saturated in HEK293T cells, and/or that NEDD4 overexpression may lead to the expression of largely inactive forms of the enzyme [21]. Overall, these results indicate that NDFIP2 is an important regulator of the steady-state levels of IFITM3 in the cell.

Given that the genetic removal of NDFIP2 in A549 cells resulted in a decrease in the steady-state levels of IFITM3, we sought to determine whether this translated into a higher susceptibility of these cells to viral infection. To this end, A549 cells were challenged with a replication-competent Vesicular Stomatitis Virus (VSV), as well as with a single round of infection-competent HIV-1 vectors pseudotyped with the VSV-G pantropic envelope protein. Both viruses carried a GFP expression cassette that allowed for the quantification of the number of infected cells using flow cytometry.

Under these conditions, we found that NDFIP2-KO cells exhibited higher susceptibility to viral challenge than control cells (Figure 2c, as indicated). Despite the fact that NDFIP2 can exert additional effects on the virus life cycle, these results along with the decreased levels of accumulation of IFITM3 support the hypothesis that NDFIP2 at least partly regulates the cell susceptibility to infection via its role in IFITM3.

### 3.3. NDFIP2 Is an Integral Part of the NEDD4-IFITM3 Complex

NDFIP2 is a well-known cellular partner of NEDD4 [21], and NEDD4 has been described to interact with IFITM3 [14]. To determine whether IFITM3 is also associated with NDFIP2, all these proteins were first ectopically expressed in pairs in HEK293T cells prior to immunoprecipitation and WB analysis. Under these conditions, IFITM3 was found to interact with NEDD4, in agreement with a previous report [14], but it was also able to interact with NDFIP2 (Figure 3a), thus validating our exosome proteomics approach to identifying novel IFITM3 cofactors. When the NEDD4:NDFIP2:IFITM3 components were expressed together, a tripartite complex could be immunoprecipitated with antibodies specific for IFITM3, NDFIP2 and NEDD4 (Figure 3b), strongly suggesting that these proteins are complexed together.

While we believe that the results obtained above support the argument of the existence of a tripartite complex, biochemical immunoprecipitations from cell lysates cannot truly distinguish this possibility from the presence of completely separated binary complexes (IFITM3:NDFIP2; IFITM3:NEDD4; NEDD4:NDFIP2) and/or from the simultaneous presence of both binary and ternary complexes. To better understand the spatial relationship between these proteins in the cell, we therefore used confocal microscopy.

When expressed individually, IFITM3 and NDFIP2 displayed a perinuclear/vesicular distribution, while NEDD4 was diffused throughout the cell cytoplasm, in agreement with the fact that NEDD4 is not a membrane-resident protein (Figure 4a). When co-expressed in a pairwise manner (Figure 4b), NDFIP2 was able to induce a clear redistribution of NEDD4 and of IFITM3, localizing them perinuclearly.

In the case of NEDD4, these results are in agreement with those of previous studies that indicate NDFIP2 as a general membrane anchor for proteins of the NEDD family [21,33]. These results are specific, because such relocalization was not observed when NDFIP2 was co-expressed with an irrelevant protein (here, the VSV-G protein, Appendix A). The co-expression of IFITM3 together with NEDD4 also led to the co-localization and redistribution of NEDD4, albeit to a lower extent than what was observed with NDFIP2. Lastly, when the three proteins were co-expressed together, significant clustering was observed between NEDD4-, IFITM3- and NDFIP2-containing compartments (Figure 4c,d for the Pearson’s coefficient correlations).

Altogether, the immunoprecipitations and the confocal microscopy analyses presented here concur in supporting the hypothesis that NDFIP2 acts as a clustering partner for both NEDD4 and IFITM3.

### 3.4. NDFIP2 Promotes the Accrued Accumulation of IFITM3 in Lysosomal Compartments

Given that NDFIP2 was the major determinant of the clustering of the complex, we sought to characterize the main subcellular region of NDFIP2-IFITM3 co-localization, using GM130 and LAMP2, markers of the cis-Golgi and lysosomal compartments, respectively. As expected from previous reports, when expressed individually, both NDFIP2 and IFITM3 were detected in significant amounts in both compartments (Figure 5a). However, when IFITM3 was expressed with NDFIP2, a higher proportion of IFITM3 was found in LAMP2-positive lysosomal vesicles, indicating that NDFIP2 promoted partial IFITM3 redistribution into this compartment (Figure 5b,c for the Mander’s coefficient correlations).

### 3.5. NDFIP2 Overexpression Leads to Accrued IFITM3 Levels in a PPxY-Dependent Manner

Our results indicate that NDFIP2 is important for recruiting NEDD4 and IFITM3 in lysosomal compartments, but they also indicate that this may oppose a more efficient degradation of IFITM3 by NEDD4. NDFIP2 possess three proline-rich motifs that are associated with the four tryptophan–tryptophan (WW) domains present in NEDD4 [21]. To determine whether the ability of NDFIP2 to titer out NEDD4 is dependent on these domains, we compared the effects of wild-type NDFIP2 to those of an NDFIP2 mutant in which the three PPxY domains had been mutated (named PY1,2,3/F, in which the tyrosine residues of each domain were mutated to phenylalanine [21], Figure 6a). Under these conditions and as observed above, the co-expression of IFITM3 and WT NDFIP2 in HEK293T cells led to an increase in the steady-state levels of IFITM3. However, such an increase was essentially lost in the presence of the NDFIP2 PY1,2,3/F mutant, supporting the notion that the action of NDFIP2 on IFITM3 requires the interaction of its PPxY domains with the WW domains of NEDD4.

To further support our model, we measured the extent of the ubiquitination of IFITM3 in the presence of WT or mutated NDFIP2. To do so, HEK293T cells were transfected, as described above, together with DNA coding for an HA-tagged version of ubiquitin. The cells were then lysed prior to the immunoprecipitation of IFITM3 and the WB analysis of the extent of ubiquitination using an anti-HA antibody (Figure 6b). Under these conditions, we found that the extent of IFITM3 ubiquitination decreased in the presence of the wild-type but not the PY1,2,3/F NDFIP2 mutant, overall supporting our model in which NDFIP2 protects IFITM3 from ubiquitin-mediated degradation by NEDD4.

## 4. Discussion

Using a proteomics-based approach, we report here the identification of NDFIP2 as an important regulator of the steady-state levels of IFITM3. Our results indicate that NDFIP2 acts as an independent clustering factor for both NEDD4 and IFITM3 on lysosomal vesicles, where it leads to a positive regulation of IFITM3, likely by competing locally with IFITM3 for NEDD4. In support of the hypothesis that NDFIP2 can divert NEDD4 away from its IFITM3 target, the removal of NDFIP2 leads to a decrease in the intracellular levels of IFITM3, while its overexpression leads to a specular increase that requires the PY modules of NDFIP2, which are known to engage the corresponding WW domains of NEDD4. Finally, this correlates with the extent of ubiquitin accumulation on IFITM3. With the caveat that NDFIP2 is very likely to control the levels of different proteins in addition to IFITM3, the fact that the removal of NDFIP2 leads to a decrease in IFITM3 levels and that it concomitantly exposes the cell to infection is compelling and in line with the broad antiviral role of IFITMs.

NDFIP2 is a short-lived transmembrane protein that resides in the Golgi and in multivesicular bodies [33], and it serves as a membrane anchor for several HECT E3 ubiquitin ligases, such as NEDD4, NEDD4L, Itch, SMURF1 and SMURF2 [21]. Together with NDFIP1, with which it shares 52% homology, NDFIP2 therefore acts as a general membrane adaptor for E3 ubiquitin ligases that are otherwise cytoplasmic. The NDFIP:HECT member interaction engages the PY modules present on NDFIPs with the correspondent WW ones present on HECT members. In the case of NEDD4, recruitment by NDFIP2 can lead to an increase in the degradation rates of several NEDD4 substrates, such as the Janus kinase 1 and 2 (JAK1/2); the Roundabout homolog 1 protein (Robo); the phosphatase and tensin homolog tumor suppressor (PTEN); the divalent metal transporter (DMT1); or diverse members of the Src family. However, NDFIP2 can also drive an opposite behavior and lead to the stabilization of NEDD4 targets, such as the mature form of the Ether-a-go-go-Related Gene (hERG) or Connexin43 [25,34,35,36,37,38,39].

The outcome of the action of NDFIP2 on the complex between NEDD4 and its substrates is therefore variable and cannot be predicted a priori, likely reflecting the dynamic nature of the different complexes that can be formed between NDFIP2, NEDD4 and each of their substrates. In the case of IFITM3, our data indicate that IFITM3 can bind directly to NDFIP2, as well as NEDD4, similarly to NEDD4, which can also bind both components individually. This likely favors the co-existence of binary and ternary complexes that concur in influencing the extent of accumulation of IFITM3.

In this respect, NDFIP2 may be an important modulator of the interaction between NEDD4 and IFITM3. NDFIP2 has been cloned as a gene exquisitely upregulated during T-cell activation and downregulated in T cells treated with Cyclosporin A (CsA) [20], and its expression has been linked to the polarization of T cells towards an inflammatory phenotype (Th1) [40]. As such, these results suggest that NDFIP2 is responsive to external cues, and, given its important role in the regulation of the interaction between HECT members and their targets, it may represent an important hub to translate external stimuli into physiological changes. It is of interest that NDFIP2 is phosphorylated by members of the Src kinase, which are themselves not only targets of NEDD4-mediated degradation but also regulators of the activity of IFITM3 via phosphorylation. At present, the influence of phosphorylation on the relationship between NDFIP2, NEDD4 and IFITM3 is unclear, but it certainly represents an important notion to explore. As such, NDFIP2 levels may represent a fine-tuning step in the regulation of the steady-state levels of IFITM3, leading to dynamic changes in the cell susceptibility to viral infection. A number of studies have led to the identification of several protein cofactors and post-translational modifications of IFITMs, advancing our knowledge of the biology of this antiviral factor. Yet, our comprehension of how cofactors and post-translational modifications influence each other and concur to the overall regulation of IFITM molecules still remains superficial. For example, it has been shown that the phosphorylation of Tyr 20 can alter the extent of IFITM3 ubiquitination on lysine residues. However, we ignored whether ubiquitination competes with methylation, another post-translational modification that also occurs on lysine residues. Similarly, it is unclear whether the extent of IFITM phosphorylation is itself tributary of the activation status of kinases of the Src family, which is in itself a very complex topic. As such, we believe that future efforts in the field will likely be directed towards providing a more complex picture of the relationships that reported cofactors and post-translational modifications bear in the physiology of IFITM proteins.

Finally, NDFIP2 gene expression has been shown to be significantly decreased in acute myeloid leukemia (AML) patients in a study that reported a strong correlation between high ifitm3 expression and adverse prognosis in infected AML patients [41]. It would be therefore tempting to speculate that the overall increase in IFITM3 expression in these AML patients is directly linked to NDFIP2 downregulation, similarly to what was observed in this study.

## 5. Conclusions

Overall, our study indicates NDFIP2 as an important regulator of IFITM3 and as an important contributor to the biology of this antiviral factor. In light of the exquisite regulation of this factor and of its response to external cues, NDFIP2 may be one of the manners through which the cell finely tunes its IFITM levels and thus alters its susceptibility to viral challenge.

## Figures and Tables

**Figure 1 viruses-15-01993-f001:**
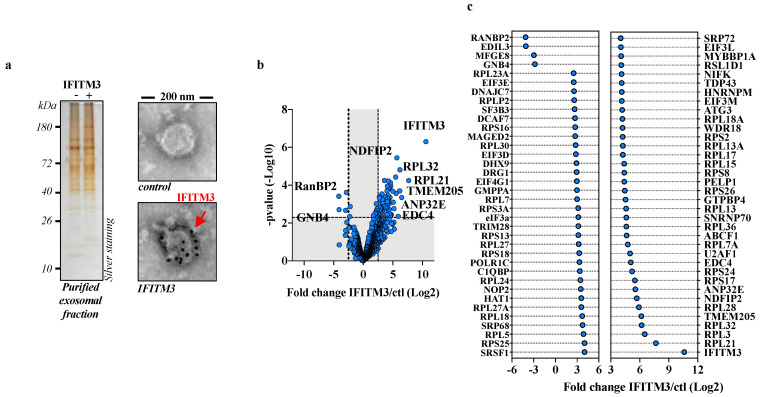
MS-based proteomic analysis of exosomal vesicles obtained in the presence or absence of IFITM3. (**a**) HEK293T cells were transfected with DNAs coding for Flag-IFITM3 or control DNA. Twenty-four hours after media change, cell supernatants were harvested, syringe-filtered and purified first using ultracentrifugation through a 45/25% sucrose cushion. The interface between the two sucrose steps was harvested, diluted three times and subjected to a second ultracentrifugation through a 25% sucrose cushion. Pellets were then resuspended, normalized by protein content and analyzed using SDS-PAGE/silver staining or using immuno-gold electron microscopy with an antibody specific for the N-terminal Flag tag of IFITM3. (**b**) Volcano plot displaying the differential abundance of proteins in vesicles from cells expressing or not Flag-IFITM3 and analyzed using MS-based label-free quantitative proteomics. Three independent replicates were analyzed. The volcano plot represents the -log10 (limma *p*-value) on y axis plotted against the log2(fold change IFITM3 vs. Ctl) on x axis for each quantified protein. Blue dots represent proteins found differentially abundant (log2(fold change) ≥ 2.5 or ≤ −2.5 and limma *p*-value < 0.005, leading to a Benjamini–Hochberg FDR < 1%). (**c**) Graph presenting more specifically proteins significantly enriched/decreased in the presence of IFITM3 over control (log2(fold change) ≥2.5 or ≤−2.5, limma *p*-value < 0.005).

**Figure 2 viruses-15-01993-f002:**
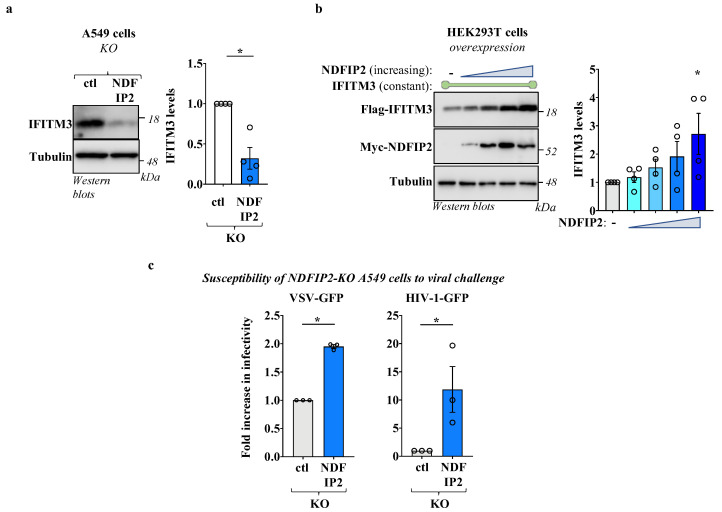
NDFIP2 modulates IFITM3 levels, and NDFIP2-KO cells exhibit increased susceptibility to viral infection. (**a**) Stable A549 cells genetically knocked out for NDFIP2 or control were generated using lentiviral-mediated CRISPR/Cas9 gene delivery and deletion. Stable cells were selected as pools thanks to the puromycin resistance gene carried by the vector and then analyzed using WB. The quantification of the IFITM3 levels obtained after densitometric quantification is shown in the graph (n = 4; Avg and SEM; *, *p* = 0.0025, following an unpaired, two-tailed, Student *t*-test between the two conditions). (**b**) HEK293T cells were transfected with a constant amount of DNA coding for Flag-IFITM3 together with increasing doses of DNA coding for either myc-NDFIP2 (0.1 μg of IFITM3 and 0.05 to 0.4 μg of NDFIP2 in 24-well plates), prior to WB analysis twenty-four hours post-transfection. As in (**a**), the WB panels present typical results, while the graphs present the densitometric quantification of IFITM3 signals obtained after WB (n = 4; * and ns indicate *p* < 0.05 and non-significance, respectively, according to one-way Anova with Dunnett’s multiple comparison tests between the indicated conditions). (**c**) Pools of control or of NDFIP2 knockout A549 cells were used as target for viral challenge with a multiplicity of infection (MOI) of 0.5 for single round of infection-competent HIV-1 vectors, pseudotyped with the pantropic envelope VSV-G to allow their entry into A549 cells, and with an MOI of 0.01 for replication-competent VSV virus. Given that both carried the GFP reporter gene, viral infection was assessed two to three days after viral challenge using flow cytometry in the case of HIV-1 and 16 h in the case of VSV. The graphs of viral infection present AVG and SEM of three independent experiments. *, *p* < 0.05, following a Student *t*-test between the indicated conditions (unpaired, two-tailed).

**Figure 3 viruses-15-01993-f003:**
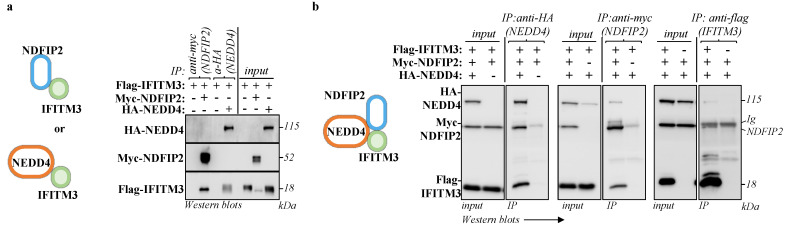
NDFIP2 is part of the NEDD4-IFITM3 complex. HEK293T cells were transfected with DNAs coding for NDFIP2, IFITM3 and NEDD4, in pairwise combinations (**a**) or together as indicated (**b**), prior to cell lysis and immunoprecipitations with the indicated antibodies twenty-four hours later. The panels present typical results obtained from four independent experiments.

**Figure 4 viruses-15-01993-f004:**
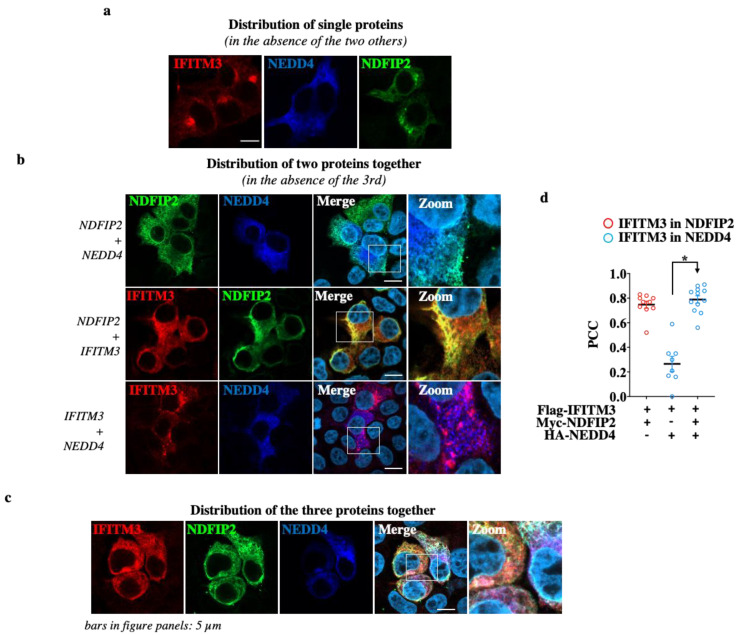
NDFIP2 heightens NEDD4 recruitment on IFITM3. HEK293T cells were transfected with DNAs coding for NDFIP2, IFITM3 and NEDD4, alone (**a**), in pairs (**b**) and all together (**c**), as indicated, prior to confocal microscopy analysis twenty-four hours later. Representative pictures are presented here. (**d**) Graph presenting the Pearson’s correlation coefficients (PCCs) among the indicated proteins calculated from 10 to 12 cells. * *p* < 0.0001, following a Student *t*-test among the indicated conditions (unpaired, two-tailed).

**Figure 5 viruses-15-01993-f005:**
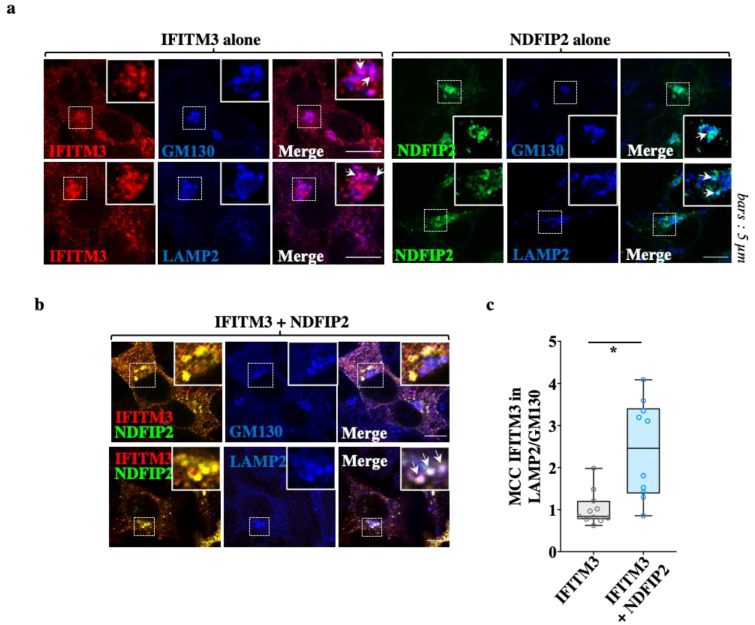
NDFIP2 leads to an accrued lysosomal redistribution of IFITM3. (**a**,**b**) HEK293T cells transfected as above were analyzed using confocal microscopy together with markers specific for the cis-Golgi and for lysosomes (GM130 and LAMP2, respectively). (**c**) The relative distribution of IFITM3 in each compartment is shown as a ratio of the Mander’s correlation coefficients (10 to 11 cells from three independent experiments) in the Box and Whisker plot. *p* = 0.0012, following a Student *t*-test among the indicated conditions (unpaired, two-tailed).

**Figure 6 viruses-15-01993-f006:**
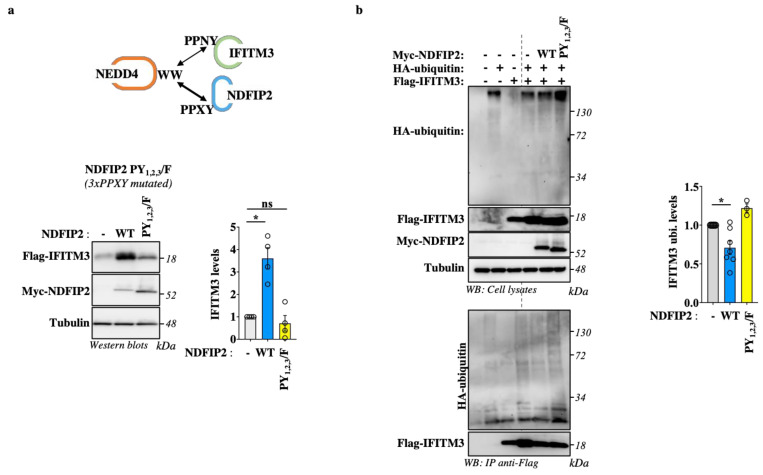
NDFIP2 overexpression leads to accrued accumulation of IFITM3 in a PPxY-dependent manner and decreases the extent of IFITM3 ubiquitination. (**a**) HEK293T cells were transfected with DNAs coding for Flag-IFITM3 together with wild-type and PY_1,2,3_/F NDFIP2 mutant (in which the key tyrosine residue in each of the three PPxY domains in NDFIP2 had been mutated to phenylalanine), prior to WB analysis twenty-four hours post-transfection. The WB panels present typical results obtained (n = 4), while the graphs present the densitometric quantification of IFITM3 signals obtained after WB (n = 4; * and ns indicate *p* < 0.05 and non-significance, respectively, according to one-way Anova with Dunnett’s multiple comparison tests between the indicated conditions). (**b**) HEK293T cells were similarly transfected together with an HA-tagged version of ubiquitin. Cells were lysed, and the extent of IFITM3 ubiquitination was assessed via immunoprecipitation of IFITM3 and WB analysis, using an anti-HA antibody. The panels present typical results obtained, while the graph present the densitometric quantification of ubiquitin smears (Avg and SEM; n = 3 to 7 independent experiments; *, *p* < 0.05 according to one-way Anova with Tukey’s multiple comparison tests between the indicated conditions).

## Data Availability

Source data are provided with this paper. There is no restriction on data availability.

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
