# Peer review of "A Proteomics-Based Approach Identifies the NEDD4 Adaptor NDFIP2 as an Important Regulator of Ifitm3 Levels"

_viruses, 2023, doi:10.3390/v15101993_

Round 1

Reviewer 1 Report

Here, the authors designed an elegant approach to identify IFITM3 interacting proteins through proteomic analysis of exosomes that contain high levels of IFITM3. They reasoned that IFITM3 interacting partners should also be present in extracellular vesicles. Among protein hits, the authors identified NDFIP2 and provided evidence that this protein interacts with IFITM3 and regulates the IFITM3 levels. It was found that overexpression or knockout of NDFIP2 increased or reduced the IFITM3 level in cells, respectively. Accordingly, NDFIP2 knockout increased viral infection. NDFIP2 is transmembrane protein that serves as an adaptor of NEDD4 E3 ubiquitin ligases which are known to regulate the IFITM3 levels. The results suggest that NDFIP2 binds to both IFITM3 and NEDD4 and that these complexes are targeted to lysosomes. The authors propose that NDFIP2 prevents IFITM3 ubiquitination by NEDD4 and thereby prevents its degradation.

 Collectively, this study identifies and validates a novel IFITM3 interacting partner, NDFIP2. NDFIP2 which regulates the cellular level of IFITM3, apparently by forming a ternary NDFIP2-IFITM3-NEDD4 complex and thereby suppressing IFITM3 degradation. This model is supported by experimental results, primarily co-IP of these proteins. The imaging results showing colocalization of these proteins in cells, although less convincing due to poor quality of images (see below), appear to support the main conclusion.

 Images in Figure 4 are difficult to interpret. Please show higher resolution and larger images to better represent subcellular localizations of proteins of interest. From what can be gleaned from these images, the baseline IFITM3 distribution in HEK293T cells does not conform with the traditional endosomal (punctate) appearance and is rather “reticulated/fibrous”. This contrasts the IFITM3 staining pattern in Figure 5. The triple colocalization image (Figure 4, bottom) also features fibrous IFITM3 staining pattern. Here focusing on the perinuclear bright triple positive spot (likely the Golgi) does not do justice, as non-interacting proteins could be present in the Golgi when overexpressed. It might be worth testing colocalization of these proteins using cells endogenously expressing all three proteins.

Minor points:

 1.     Please comment on the reason a Mander’s coefficient was used to quantify colocalization in Figure 5, whereas Person’s coefficient was used to analyze all other images.

2.  Please delete lines 446-448.

Reviewer 2 Report

Marziali and colleagues take a clever approach towards identifying protein regulators of the broadly antiviral protein IFITM3, leveraging past knowledge of IFITM3's packaging into extracellular vesicles. Proteomic analysis of IFITM3-containing vesicles revealed differential protein levels compared to a vector control.  A top enriched hit, NDFIP2, was chosen as the leading candidate to validate due to its association with the known negative regulator of IFITM3, NEDD4.  NDFIP2 KO decreases endogenous IFITM3 expression, increasing susceptibility of these cells to Vesicular Stomatitis Virus infection.  Contrastingly, over expression of NDFIP2 increases exogenous IFITM3 expression.  The authors perform a range of well-controlled biochemical and imaging experiments to suggest that NDFIP2 exists in complex with IFITM3 and NEDD4, protecting IFITM3 from NEDD4-mediated degradation.  Further, protection of IFITM3 requires PPXY motifs present in NDFIP2 that mediate interaction with NEDD4.  The results are compelling, and reveal a novel layer of IFITM3 regulation that may be leveraged in the future towards IFITM-mediated antiviral strategies.  However, the authors make some assumptions that I feel need to be experimentally confirmed prior to publication of this study.

My major concerns are as follows:

1.  I was unable to find confirmation of NDFIP2 knockout in the A549 pooled cells.  With the high homology with NDFIP1, can this be used as a control for KO specificity? It's possible that there is no reliable endogenous antibody of NDFIP2.  However, PCR validation of cDNA, and/or qPCR quantification of verified NDFIP2-specific probes would also be an acceptable proxy for KO validation.  I don't consider the IFITM3 phenotype sufficient to ensure these cells are specifically knocked out.

2.  The assumption is that NDFIP2 protects IFITM3 from degradative ubiquitination by NEDD4, though this is not directly shown.  I believe the authors have the tools and the technical prowess to evaluate IFITM3 ubiquitination in the presence of WT NDFIP2 or PY123/F NDFIP2 (an expansion of Figure 6).  Alternatively, or in addition, endogenous IFITM3 ubiquitination could be measured in pooled NDFIP2 KO cells following validation of KO.

3.  Figure 7, and the corresponding section, feel incomplete and detract from the mechanistic discoveries in the preceding figures.  It is well documented that A549 cells with less IFITM3 will be more susceptible to VSV and other IFITM3-sensitive viruses.  Furthermore, line 430, mentioning the susceptibility of two different viruses families is misleading, as the VSV-G pseudotyped lentivirus will behave akin to wild-type VSV where IFITM3 is concerned.  A replication-competent influenza infection would be more appropriate here, though the wild-type VSV infection data is sufficient if this data and the corresponding section are placed in the final panel of Figure 2.

4.  Densitometric quantification should be added to methods.

Minor concerns:

Line 55-57: Discussion of interdependence of post-translational modifications does not mention the overlap between phosphorylation and ubiquitination discussed in ref 13.  This overlap is highly pertinent to this study and would be a good follow-up experiment to propose in the discussion.

Line 79: "by the Yount Lab"

Line 192: "1 ug" should be lowercase

Line 268-272: Is it possible that translation processes are enriched in IFITM3-expressing cells because an empty vector was used as a control?  It's possible that a transgene like GFP would have enriched these same genes.

Figure 3B:  Brackets above input and IP blots should only be above the IP, as no IP is performed for the input blots.

Figure 5: Are there error bars for this experiment?

Figure 5 and 6 should be divided into separate panels.

Line 446-448: Remove

Round 2

Reviewer 2 Report

The authors have carefully considered the comments from both reviewers and have added new evidence that greatly improves the manuscript.  I feel that this study is complete and will be of great interest to the readers of Viruses.  I have no other concerns.